# Evaluation of the Mycobacterium RealTime Kit Vircell (MRTVircell) Assay for Detecting *Mycobacterium* Species in Clinical Specimens

**DOI:** 10.3390/pathogens14050429

**Published:** 2025-04-28

**Authors:** Maria Aguilera Franco, Antonio Sampedro Padilla, Juan Francisco Gutiérrez-Bautista, Adrián González Martínez, Fernando Cobo, José Gutiérrez-Fernández, Juan Antonio Reguera, Jose María Navarro Mari, Javier Rodríguez-Granger

**Affiliations:** 1Department of Microbiology, Virgen de las Nieves University Hospital, 18014 Granada, Spain; maria.aguilera.franco.sspa@juntadeandalucia.es (M.A.F.); adrian.gonzalez.martinez.sspa@juntadeandalucia.es (A.G.M.); fernando.cobo.sspa@juntadeandalucia.es (F.C.); josegf@ugr.es (J.G.-F.); jantonio.reguera.sspa@juntadeandalucia.es (J.A.R.); josem.navarro.sspa@juntadeandalucia.es (J.M.N.M.); 2Biosanitary Research Institute of Granada, 18012 Granada, Spain; juanfry@ugr.es; 3Zaidín Sur Health Center, Andalusian Health Service, Granada Metropolitan District, 18007 Granada, Spain; 4Department of Clinical Analysis and Immunology, Virgen de las Nieves University Hospital, 18014 Granada, Spain; 5Department of Microbiology, School of Medicine, University of Granada, 18016 Granada, Spain

**Keywords:** *Mycobacterium tuberculosis* complex, nontuberculous mycobacteria, evaluation, real-time PCR, *Mycobacterium avium* complex, *Mycobacterium abscessus* complex

## Abstract

Rapid and accurate diagnosis of mycobacterial infections is crucial for guiding therapeutic decisions. This study presents the first evaluation of a novel molecular assay, the Mycobacterium RealTime PCR Kit Vircell (MRTVircell), a real-time PCR-based test designed for the specific detection of *Mycobacterium tuberculosis* complex (MTBC), *Mycobacterium avium* complex (MAC), *Mycobacterium abscessus* complex (MABC), and other nontuberculous mycobacteria (NTM) in both respiratory and non-respiratory samples. The evaluation was conducted under routine workflow conditions using 721 clinical specimens, including 559 respiratory and 162 non-respiratory samples. Among these, 5.69% were smear-positive, 6.38% were culture-positive for MTBC, and 9.85% were culture-positive for NTM. The performance of the MRTVircell was compared to both culture results and the Anyplex MTB/NTM real-time PCR assay. The two PCR systems demonstrated a 96.95% overall concordance rate for the detection of MTBC, NTM, and negative specimens. Based on culture as the reference method, the sensitivity, specificity, positive predictive value, and negative predictive value of the MRTVircell for MTBC detection were 80.43%, 99.64%, 94.87%, and 98.41%, respectively, while for Anyplex MTB/NTM (Seegene), these values were 76.09%, 99.64%, 94.59%, and 98.06%, respectively. For NTM detection, the sensitivity, specificity, positive predictive value, and negative predictive values were 28.17%, 99.29%, 83.33% and 91.63% for MRTVircell and 21.13%, 99.11%, 75%, and 91.67% for Anyplex MTB/NTM, respectively. MRTVircell is a rapid and reliable tool for the detection and differentiation of MTBC, MAC, MABC, and other NTM in clinical samples.

## 1. Introduction

The genus *Mycobacterium* comprises over 200 species [1]. Mycobacterial diseases represent a significant global health concern [2].

It is estimated that in 2023, 10.8 million people worldwide contracted tuberculosis (TB), resulting in 1.25 million deaths. TB has likely once again become the leading cause of death from an infectious pathogen globally. Moreover, in recent years, nontuberculous mycobacteria (NTM) have emerged as an increasing cause of disease in various regions worldwide [3].

NTM refer to all mycobacterial species except those belonging to the *Mycobacterium tuberculosis* complex (MBTC) and *Mycobacterium leprae*. These microorganisms are widely distributed in the environment and are frequently detected in water and soil. Although the true prevalence of NTM diseases remains unknown in most countries, there is growing evidence that their prevalence is rising globally [2].

NTM infections often present with symptoms similar to TB, as the lungs are the primary site of infection. However, their lower specificity increases the risk of misdiagnosis, potentially leading to adverse patient outcomes. Therefore, rapid and accurate pathogen identification is crucial for precise diagnosis, effective treatment, and proper infection control of both MBTC and NTM infections [2,4,5].

The microbiological diagnosis of mycobacteria has traditionally relied on smear microscopy of clinical samples and cultures. The detection of acid-fast bacilli (AFB) using microscopic examination (Ziehl–Neelsen and/or auramine–rhodamine staining) is the quickest, easiest, and cheapest method available; however, it has limited sensitivity, especially in geographical areas of lower incidence, in extrapulmonary forms (paucibacillary) and in patients with HIV, and does not differentiate between MTBC and NTM [6,7]. For optimal recovery of *Mycobacterium* spp., a combination of broth and solid media should be used. Various types of broth media are available, many of which are incorporated into automated mycobacterial detection systems. Solid media are also included to ensure the recovery of rare strains that may not grow in broth [6].

For species-level identification, biochemical tests and high-performance liquid chromatography have traditionally been used, but these methods have gradually been replaced by mass spectrometry (MALDI-TOF) and molecular techniques. However, these techniques are primarily performed on isolates obtained through culture, which remains the current gold standard. As a result, identification, particularly of slow-growing mycobacteria, can be significantly delayed [8,9].

Therefore, in recent years, various strategies have been proposed to achieve rapid diagnosis not only of active TB but also of other non-tuberculous mycobacteria involved in disease. These strategies are diverse, focusing on improving conventional techniques and incorporating genotypic, proteomic, and even bacteriophage-based methods. Among these, nucleic acid amplification techniques currently have the most practical utility, although they still present certain limitations when compared to culture [10].

Over the past decade, new molecular methods based on nucleic acid amplification tests (NAATs) have been developed to improve diagnostic sensitivity [7]. Several commercial molecular tests for detecting MTBC are already recommended by the World Health Organization (WHO) and are widely available. NAATs can identify MTBC and detect mutations in resistance genes associated with key anti-tuberculosis drugs, such as rifampin and isoniazid [11]. Additionally, advances in molecular diagnostics have also transformed the identification of NTM, providing significant improvements over traditional methods in terms of accuracy, sensitivity, speed, and cost-effectiveness [12].

Several commercial real-time multiplex PCR kits have been developed for the rapid molecular detection and differentiation of MTBC and NTM. These kits have been evaluated using clinical respiratory specimens, particularly from culture sample positives [13,14].

Despite the advances in molecular diagnostic tools for mycobacteria, these methods do not replace traditional diagnostic tests but rather complement the established diagnostic approach. This is a topic of ongoing debate due to the high cost and the need for specialized laboratories and qualified personnel. Additionally, PCR-based techniques do not differentiate between viable and non-viable microorganisms, which makes them less useful for patients undergoing treatment, as they may detect residual DNA from non-viable organisms. Furthermore, in areas with low prevalence of disease, the positive predictive value of these molecular methods may be limited, potentially leading to false positives. In these contexts, traditional methods such as culture and AFB stain remain essential for confirming the presence of viable organisms and providing accurate diagnostic results [10].

The primary aim of our study was to evaluate the Mycobacterium RealTime PCR Kit Vircell (MRTVircell), which is designed with specific genetic targets to differentiate major Mycobacteria species. This kit includes distinct primers for the detection of the MTBC, *Mycobacterium avium* complex (MAC), *Mycobacterium abscessus* complex (MABSC), and NTM. The results were compared with those obtained from culture and another real-time PCR assay, Anyplex MTB/NTM (Seegene).

## 2. Materials and Methods

### 2.1. Clinical Samples, Acid-Fast Bacilli (AFB) Smear, and Mycobacterial Culture

We collected clinical samples from patients referred for mycobacterial studies between June and October 2023 at the Microbiology Laboratory of Hospital Universitario Virgen de las Nieves. Our center is located in the Mediterranean coastal region of southeastern Andalusia and serves the entire provincial population suspected of having mycobacterial infections.

The mean annual rate of respiratory TB in Andalusia was between 6.41 and 5.34 per 100,000 inhabitants in 2023. The true incidence of other mycobacterial infections remains unknown, as nontuberculous mycobacteria are not classified as reportable diseases in Spain [15].

The original samples used in the evaluation comprised a total of 721 specimens, including 560 respiratory samples [427 (76.25%) sputum, 96 (17.14%) bronchoalveolar lavage, and 37 (6.6%) bronchial aspirates] and 161 non-respiratory samples [82 (50.93%) pleural fluid, 29 (18.2%) biopsies from various sites, 15 (9.32%) abscesses from different locations, 15 (9.32%) urine samples, 14 (8.7%) other sterile body fluids, and 4 (2.48%) bone marrow aspirates, 2 (1.24%) gastric lavages]. No duplicate samples were included in the analysis.

For all clinical samples, an acid-fast bacillus (AFB) smear was performed using auramine–rhodamine fluorescent staining, followed by confirmation with Ziehl–Neelsen staining in positive cases. Mycobacterial culture was conducted using liquid media BD Bactec^TM^; MGIT^TM^ (Becton Dickinson, Sparks, MD, USA) and/or VersaTREK^TM^ (Thermo Fisher Scientific^TM^, Waltham, MA, USA), as well as solid culture (Löwenstein-Jensen Pyruvate Medium, RPD Microbiology, Barcelona, Spain). Cultures were maintained for six weeks following the decontamination of non-sterile samples, in accordance with standard protocols [6,16]. PCR studies were performed using 1 mL aliquots from each sample, either directly or after decontamination. The aliquots were stored at 4 °C until testing, which was conducted within 48 h of sample processing. Molecular assays were performed either upon request by the clinician responsible for the patient or when our laboratory identified clinical, radiological, or epidemiological factors suggesting that an early molecular diagnosis could be beneficial. This approach ensured that the study population encompassed cases where PCR-based diagnosis was deemed clinically relevant.

Identification of different mycobacterial complex members was performed in liquid medium using the commercial GenoType Mycobacterium CM/AS kit (Hain LifeScience, Nehren, Germany) and species-level identification for MAC and MABS was conducted using the GenoType NTM-DR kit (Hain Lifescience, Nehren, Germany). Subcultures were performed on solid media (NTM elite agar and blood agar, bioMérieux, Marcy-l’Étoile, France) and species identification was confirmed by MALDI-TOF mass spectrometry (Bruker, Bremen, Germany) from growth on solid media.

### 2.2. Mycobacterium RealTime PCR Kit Vircell Assay (MRTVircell)

This method is based on the amplification, within the same reaction well, of specific nucleic acid fragments from mycobacteria belonging to the MTBC, MAC, and MABSC, as well as NTM, using real-time PCR. The PCR mix targets a specific fragment of the IS1081 insertion sequence for MTBC, the ITS region for MAC and MABSC, and the 16S rRNA gene for the *Mycobacterium* genus. As a control procedure, the kit amplifies the human RNAse P gene as an internal control to ensure proper sample extraction, the absence of amplification inhibitors, and the correct assay setup.

Prior to extraction, 1 mL of each sample was inactivated by incubation at 100 °C for 20 min, followed by centrifugation. The procedure consists of two main steps: nucleic acid extraction and PCR setup. Nucleic acid extraction was fully automated using the MagXtract 3200 System (Chroma ATE Inc., Taoyuan, Taiwan), following the manufacturer’s protocol. Briefly, 600 μL of the processed sample was added to the extraction system, where nucleic acids were purified using the TANBead Nucleic Acid Extraction Kit (Taiwan Advanced Nanotech Inc., Taoyuan, Taiwan). The extraction process involved automated lysis, binding to nanomagnetic beads, multiple wash steps, and final elution in 50 μL of buffer. For PCR setup, 5 μL of extracted nucleic acid was added to the reaction tube containing the reagents provided in the MRTVircell assay. PCR amplification was performed on the CFX96 Real-Time PCR Detection System (Bio-Rad Laboratories, Hercules, CA, USA). Each run included positive and negative controls supplied in the kit to ensure the accuracy of the amplification. Automated result interpretation was conducted using Vircom middleware (Vircell SL, Granada, Spain). The threshold cycle (Ct) values used to determine positive results in the MRTVircell assay were <37 for NTM and <40 for MTBC, MAC, and MABS, according to the manufacturer’s recommendations.

### 2.3. Anyplex MTB/NTM (Seegene) Real-Time PCR Assay

The Anyplex MTB/NTM assay and sample extraction were performed in parallel with the MRTVircell assay, using the CFX96 Real-Time PCR Detection System thermal cycler (Bio-Rad Laboratories, Hercules, CA, USA), following the manufacturer’s instructions. After decontamination, DNA extraction solution (provided in the kit) was added to the clinical samples. The samples were then heated at 100 °C for 20 min, centrifuged at 15,000× *g* for 5 min, and 5 µL of the supernatant was used as the PCR template. This was mixed with 15 µL of Anyplex PCR master mix. Each run included positive and negative controls to ensure the accuracy of the amplification process, along with plasmid DNA as an internal control to verify consistent amplification of the internal control, MTBC, and mycobacterial target DNA. Results were automatically interpreted using the instrument’s software based on the manufacturer’s thresholds [17].

### 2.4. Data Analysis

Data were collected and organized using Microsoft Excel. The sensitivity, specificity, positive predictive value (PPV), and negative predictive value (NPV) of the molecular assays were evaluated and calculated using SPSS Statistics, version 27 (IBM, Armonk, NY, USA). Cohen’s kappa coefficient was used to assess the level of agreement between methods. McNemar’s test was applied to compare the sensitivity of the two methods.

## 3. Results

The specimens were collected from 626 patients, 288 (46.2%) females and 338 (53.8%) males with an age range of 1–98 years old (mean 52.03) +/− SD 17.71.

Among the 721 samples submitted for culture, 117 (16.23%) tested positive for either MTBC or NTM. Of these, 46 (39.32%) were positive for MTBC, with 43 (93.48%) being respiratory and 3 (6.52%) non-respiratory. The remaining 71 (60.68%) were positive for NTM, including 64 (90.14%) respiratory and 7 (9.86%) non-respiratory samples. Of the 71 NTM strains recovered, the three most prevalent species were primarily from the MAC: *M. avium* (n = 17; 23.94%), *M. intracellulare* (n = 14; 19.72%), and *M. chimaera* (n = 4; 5.63%). Other common NTMs included *M. chelonae* (n = 14; 19.72%) and *M. gordonae* (n = 8; 11.27%), followed by *M. abscessus* (n = 3; 4.23%) and *M. lentiflavum* (n = 3; 4.23%). The remaining NTMs were isolated in single samples and identified as *M. arupense*, *M. fortuitum*, *M. kansasii*, *M. kumamotonense*, *M. mageritense*, and *M. mucogenicum*. No samples showed simultaneous growth of both MTB and NTM in culture.

A total of 40 samples (5.5%) were excluded from the sensitivity and specificity evaluation due to culture contaminated by bacteria or fungi. Among these, both PCR tests yielded negative results in 38 out of 40 cases (95%), while two samples (5%) tested positive for NTM with both MRVircell and Anyplex MTB/NTM. Since no MTB cases were affected, and the two NTM-positive cases yielded concordant results, these exclusions are unlikely to have significantly impacted the reported sensitivity and specificity values. Nonetheless, culture contamination remains an inherent limitation in mycobacterial diagnostics.

The detection rates for MTBC in respiratory and non-respiratory samples according to each molecular diagnostic kit were as follows: 38/43 (88.37%) and 1/3 (33.33%), respectively, for MRTVircell; and 36/43 (83.72%) and 1/3 (33.33%), respectively, for Anyplex MTB/NTM (as seen in Table 1).

Based on the culture results, MRTVircell demonstrated a sensitivity of 80.43%, a specificity of 99.64%, a positive predictive value of 94.87%, and a negative predictive value of 98.41% for MTB detection. The Anyplex MTB/NTM showed corresponding values of 76.09%, 99.64%, 94.59%, and 98.06%, respectively (as seen in Table 2A).

For AFB stain-positive samples, both kits demonstrated identical sensitivity and specificity for MTB detection, at 100% and 88.89%, respectively (as seen in Table 2A). For AFB stain-negative samples, the sensitivity and specificity for detecting MTB were 57.14% and 99.82% with MRTVircell, and 47.62% and 99.82% with Anyplex MTB/NTM (as seen in Table 2A).

The MTB detection rates in respiratory and non-respiratory samples were 7.68% (43/560) and 1.86% (3/161) for culture, 6.79% (38/560) and 0.62% (1/161) for MRTVircell, and 6.43% (36/560) and 0.62% (1/161) for Anyplex MTB/NTM, respectively (as seen in Table 1).

Based on the culture results, the sensitivity, specificity, positive predictive value, and negative predictive value for detecting NTM with MRTVircell were 28.17%, 99.29%, 83.33%, and 91.63%, respectively. For Anyplex MTB/NTM, these values were 21.13%, 99.11%, 75%, and 91.67%, respectively (As seen in Table 2B).

For AFB stain-positive samples, the sensitivity and specificity for detecting NTM with MRTVircell were 83.33% and 100%, respectively, which were identical to the values for Anyplex MTB/NTM (as seen in Table 2B).

For NTM, MRT Vircell identified 18/560 (3.21%) of respiratory samples and 9/161 (5.59%) of non-respiratory samples as positive, whereas Anyplex MTB/NTM detected 20/560 (3.57%) and 2/161 (1.24%), respectively (as seen in Table 1).

According to culture findings, the overall concordance rate between the two PCR methods was 88.11% (600/681), with a Cohen’s kappa (κ) of 0.61 for MRTVircell, indicating substantial agreement, and 0.55 for Anyplex MTB/NTM, indicating moderate agreement. Among the discordant cases, four samples were positive with culture and Anyplex MTB/NTM but negative with MRTVircell, all of which corresponded to NTM detection. Conversely, eleven samples were positive with culture and MRTVircell but negative with Anyplex MTB/NTM, including nine for NTM and two for MTB (as seen in Table 3).

Additionally, four samples were negative for culture and MRTVircell but positive with Anyplex MTB/NTM, all corresponding to NTM. Three samples tested positive with MRTVircell but negative with Anyplex MTB/NTM and culture also for NTM. Lastly, three samples were negative for both culture and Anyplex MTB/NTM but positive with MRTVircell, including two for MTB and one for NTM (as seen in Table 3).

The overall concordance rate between the two PCR methods, regardless of culture results, was 96.95% (699/721), with a Cohen’s kappa (κ) of 0.806, indicating almost perfect agreement (as seen in Table 4).

The MRTVircell enabled species-level identification in 14 samples (13 positive for MAC and 1 for MABS). Of these, six tested positive at the genus level (*Mycobacterium* spp.) with the Anyplex MTB/NTM, which does not include species-specific targets. In all cases, growth was later confirmed in liquid culture, identifying eight *M. avium*, five *M. intracellulare*, and one *M. abscessus*.

## 4. Discussion

The WHO Global Strategy for TB Prevention, Care, and Control for 2015–2035 (known as the End TB Strategy) emphasizes the importance of early TB diagnosis, highlighting the need for rapid diagnostic methods to be accessible to all individuals presenting with signs or symptoms of the disease. In line with this objective, the WHO recommends that TB programs transition from conventional microscopy as the initial diagnostic test to rapid molecular diagnostic techniques, which offer improved sensitivity and specificity [18]. The Centers for Disease Control and Prevention (CDC) advises performing nucleic acid testing (NAAT) on at least one respiratory specimen from any patient exhibiting signs and symptoms of pulmonary TB in whom a diagnosis is being considered [19].

NTM infections represent a substantial global public health concern, impacting individuals across diverse immune statuses. Recent epidemiological studies indicate an increasing incidence rate, highlighting their growing clinical relevance [20]. The rapid and accurate molecular diagnosis of NTM and their differentiation from the MTBC are crucial for the effective management of mycobacterial diseases. This distinction is particularly important, as many NTM strains exhibit resistance to antibiotics commonly used for TB treatment [21]. Consequently, these advancements contribute to more effective patient-specific treatment strategies and improved control measures for these infectious diseases.

Currently, NAATs are widely used for the rapid diagnosis of MTBC and NTM infections with variable diagnostic performance [22]. In this prospective study, we evaluate and compare, for the first time, the diagnostic performance of the multiplex real-time PCR test MRTVircell for the direct detection of *Mycobacteria* genus, MTBC, MAC, and MABC clinical specimens in a tertiary hospital in southeastern Spain.

Overall, the sensitivity and specificity of the MRTVircell assay for MTB detection were 80.43% and 99.64%, respectively, compared to 76.09% and 99.64% for the Anyplex MTB/NTM assay. The analysis included both pulmonary and extrapulmonary samples. For AFB stain-positive samples, the sensitivity and specificity of both molecular assays were identical, at 100% and 88.89%, respectively. No statistically significant differences were found between the sensitivity and specificity of the two molecular assays (McNemar’s test, *p* = 0.48).

When compared to culture, the MRTVircell assay demonstrated slightly higher sensitivity to MTBC than the Anyplex MTB/NTM assay. Although culture remains the gold standard, its prolonged turnaround time (up to six weeks) limits its utility for rapid clinical decision-making while minimizing the likelihood of false positives. We observed that the sensitivity of both assays was lower in AFB smear-negative samples, consistent with previous findings [23]. Overall, the sensitivity, particularly in AFB-positive samples, was comparable to or higher than previously reported values (70.9–86.8%) [24].

Early diagnosis of TB is essential for timely treatment and infection control. Rapid detection of MTBC and its differentiation from NTM help prevent disease transmission and ensure appropriate treatment. In this study, the MRTVircell and Anyplex assays demonstrated high sensitivity to MTB detection (80.43% and 76.09%, respectively), with MRTVircell showing slightly superior performance. This increased sensitivity may offer clinical advantages, particularly in settings where rapid and accurate diagnosis is critical for patient management and public health interventions.

For the detection of NTM, the sensitivity and specificity of the MRTVircell assay were 28.17% and 99.29%, respectively, while for the Anyplex assay, the sensitivity and specificity were 21.13% and 99.11%, respectively. There were no statistically significant differences between the two methods (McNemar’s test: sensitivity, *p* = 0.07; specificity, *p* = 0.13). For AFB smear-positive samples, the sensitivity and specificity were 83.33% and 100%, identical for both tests. Few studies have evaluated the performance of molecular tests for detecting NTM, and the overall sensitivity to NTM detection in previous studies has been lower than that for MTBC, due to paucibacillary samples or single-copy gene targets in the NAAT assays compared to multicopy targets for MTBC [22].

The high specificity (99.64% for MTBC and 99.29% for NTM) confirms the robustness of the MRTVircell assay in distinguishing true-positive cases, as well as that of the other tested kits. Despite the high specificity observed, false-negative results remain a challenge, particularly in samples with low bacterial loads. In our study, we observed that both molecular assays failed to detect some culture-positive cases, especially among non-respiratory samples. This is likely due to the paucibacillary nature of certain specimens, the presence of PCR inhibitors, or sample heterogeneity, all of which can impact molecular test performance. Additionally, the sensitivity differences between respiratory and extrapulmonary samples highlight the limitations of direct molecular detection in cases where bacterial load is low. These findings underscore the importance of using molecular tests in conjunction with other diagnostic methods, such as culture, to ensure optimal detection of mycobacterial infections [10]. Nevertheless, the high specificity values suggest that positive PCR results for NTM are reliable, although a negative result does not exclude infection. One very useful application of direct molecular detection is to rapidly differentiate MTBC from NTM in smear-positive samples.

The wide variability in pathogenicity among NTM species, ranging from low-pathogenic organisms like *Mycobacterium gordonae* to highly pathogenic species such as *Mycobacterium kansasii*, underscores the need for precise species-level identification [25]. The ability of the MRTVircell to differentiate MTBC, MAC, and MABC is particularly beneficial, as treatment regimens vary significantly between these groups and with other NTMs that may have less clinical significance. The detection of MAC in 13 samples and MABC in one sample highlights the importance of species-level identification, which is not provided by the Anyplex MTB/NTM assay. Given recent reports of likely person-to-person transmission of MABC between cystic fibrosis patients, this might not just also impact treatment but also cross-infection prevention measures. Of the 14 samples that tested positive for NTM, only 6 (42.86%) were also positive with the Anyplex MTB/NTM assay, while the remaining 8 (57.14%) were only detected by the MRTVircell assay.

The sensitivity of the MRTVircell to MTB was significantly higher in respiratory samples (80.4%) than in extrapulmonary samples (33.33%). This discrepancy aligns with previous findings, which suggest that lower bacterial loads and greater sample heterogeneity in extrapulmonary infections make molecular detection more challenging [22].

The lower MTB detection rates in extrapulmonary samples underscore the need for complementary diagnostic methods, such as culture, in suspected cases of extrapulmonary TB. In this study, sensitivity values were obtained from a diverse range of extrapulmonary samples, and the results were consistent with previous studies applying molecular tests to similar sample types [22]. For the GeneXpert Xpert MTB/RIF Ultra detection system in extrapulmonary samples, studies have reported higher sensitivity, particularly in lymph nodes, tissues, and pleural fluids. This improvement is attributed to the incorporation of two targets for MTBC detection and the system’s ability to detect bacterial loads as low as 16 CFU/mL [26].

Based on the cultures results, the concordance rate of two methods was 88.11% (600/681) and regardless of culture results was 96.95% (699/721).

Several studies have assessed the performance of Anyplex MTB/NTM; however, this is the first study to evaluate this new assay. In general, other studies have reported higher estimates than those observed in our study, which could likely be attributed to the limited number of samples and the different sample compositions used in those studies [22,23].

This study has some limitations. It included both new and previously treated cases of TB and NTM infections, which could have influenced the number and viability of mycobacteria detected with molecular testing and culture. Consequently, false-positive results due to the persistence of non-viable mycobacterial nucleic acids in treated patients could have impacted the statistical analysis.

Another limitation was the small number of positive extrapulmonary samples, which may have resulted in a lower positive predictive value. This was particularly evident for *M. abscessus* complex, where only one positive sample was identified, preventing a meaningful comparison.

Moreover, neither assay detected antibiotic resistance determinants, which may limit their clinical utility. However, in low-incidence settings like ours, where TB and drug-resistant TB cases are rare, this may be less concerning. In such contexts, given the rising incidence of NTM compared to MTBC, it may be more cost-effective to first screen for MTBC/NTM, especially if species such as MAC or MABS could be directly identified, rather than routinely testing all samples for MTBC drug resistance markers.

The MRTVircell is a CE-IVDR-marked assay for the direct detection and differentiation of *Mycobacterium* spp., MTBC, MAC, and MABC. It is provided as a lyophilized, ready-to-use master mix, offering a rapid diagnostic alternative to culture, which can take up to six weeks. The assay provides reliable results in less than 3.5 h, enabling timely clinical decision-making and potentially reducing transmission risks. In comparison to the Anyplex assay, MRTVircell differentiates MAC and MABC from other NTM species, which is crucial for guiding appropriate treatment strategies, as MAC and MABC infections require targeted therapies.

In conclusion, this study demonstrates that in a setting of low TB incidence and with the increasing prevalence of NTM infections, especially respiratory, the MRTVircell assay is a reliable and effective tool for the detection and differentiation of mycobacterial infections. Performing molecular testing with multiple mycobacterial targets enables early initiation of effective treatment, improving clinical management and patient outcomes.

## Figures and Tables

**Table 1 pathogens-14-00429-t001:** MTB and NTM positivity rates by sample type using culture and the two real-time PCR kits.

Specimen	Nº MTB Positive (%)			Nº NTM Positive (%)		
	Culture	Anyplex MTB/NTM	MRTVircell	Culture	Anyplex MTB/NTM	MRTVircell
Respiratory ^1^	43 (7.68)	36 (6.43)	38 (6.79)	64 (11.43)	20 (3.57)	18 (3.21)
Sputum	39	34	35	54	19	16
Bronchoalveolar Lavage	2	0	1	10	1	2
Bronchial aspirates	2	2	2	0	0	0
Non-respiratory ^2^	3 (1.86)	1 (0.62)	1 (0.62)	7 (4.35)	2 (1.24)	8 (4.97)
Biopsies	2	1	1	1	0	1
Sterile body fluids	1	0	0	0	0	1
Pleural fluid	-	-	-	2	2	4
Gastric lavages	-	-	-	2	0	0
Urine	-	-	-	1	0	0
Abscesses	-	-	-	1	0	2
Total ^3^	46 (6.38)	37 (5.13)	39 (5.41)	71 (9.85)	22 (3.05)	26 (3.61)

^1^ Percentages calculated based on the total number of respiratory samples (n = 560). ^2^ Percentages calculated based on the total number of non-respiratory samples (n = 161). ^3^ Percentages calculated based on the total number of samples (n = 721).

**Table 2 pathogens-14-00429-t002:** (**A**) Results of real-time PCR systems according AFB smear and culture results for MTB; (**B**)**.** Results of real-time PCR systems according AFB smear and culture results for NTM.

(**A**)								
Target	AFB Smear	Assays	Culture + (n = 117)		Culture − (564)		Sensitivity/Specificity	PPV/NPV
			PCR +	PCR −	PCR +	PCR −		
MTB	Positive	Anyplex MTB/NTM	25	0	1	8	100/88.89 [68.36–109.42]	96.15 [89.09–100]/100
		MRTVircell	25	0	1	8	100/88.89 [68.36–109.42]	96.15 [89.09–100]/100
	Negative	Anyplex MTB/NTM	10	11	1	549	47.62 [26.83–69.41]/99.82 [99.16–99.99]	90.91 [62.26–98.38]/98.04 [96.63–98.92]
		MRTVircell	12	9	1	550	57.14 [34.85–76.81]/99.82 [99.16–99.99]	92.31 [66.74–98.63]/98.39 [97.06–99.16]
	All	Anyplex MTB/NTM	35	11	2	557	76.09 [61.23–86.75]/99.64 [98.8–99.91]	94.59 [81.81–98.6]/98.06 [96.63–98.92]
		MRTVircell	37	9	2	558	80.43 [65.7–89.88]/99.64 [98.81–99.91]	94.87 [82.7–98.64]/98.41 [97.06–99.16]
(**B**)								
Target	AFB smear	Assays	Culture + (n = 117)		Culture − (564)		Sensitivity/Specificity	PPV/NPV
			PCR +	PCR −	PCR +	PCR −		
NTM	Positive	Anyplex MTB/NTM	5	1	0	8	83.33 [62.23–100]/100	100/88.89 [70.84–100]
		MRTVircell	5	1	0	8	83.33 [62.23–100]/100	100/88.89 [70.84–100]
	Negative	Anyplex MTB/NTM	10	55	5	549	15.38 [8.21–26.33]/99.1 [97.94–99.65]	66.67 [35.42–88.72]/90.89 [88.34–92.93]
		MRTVircell	15	50	4	550	23.08 [14.07–35.02]/99.28 [98.19–99.73]	78.95 [54.43–92.86]/91.67 [89.26–93.59]
	All	Anyplex MTB/NTM	15	56	5	557	21.13 [12.93–32.22]/99.11 [97.95–99.65]	75 [50.9–89.87]/90.86 [88.31–92.91]
		MRTVircell	20	51	4	558	28.17 [18.63–40.13]/99.29 [98.2–99.73]	83.33 [62.62–93.98]/91.63 [89.22–93.56]

**Table 3 pathogens-14-00429-t003:** Results of two real-time PCR kits according to culture results.

	Anyplex MTB/NTM	MRTVircell	N
Culture + (117)	+	+	46
	+	−	4
	−	+	11
	−	−	56
Culture − (564)	+	+	3
	+	−	4
	−	+	3
	−	−	554

“+” indicates detection of MTB or NTM; “−” indicates no detection of MTB or NTM.

**Table 4 pathogens-14-00429-t004:** Correlation of results between the analyzed real-time PCR kits.

		MRTVircell			
		MTB	NTM	Negative	Total
Anyplex MTB/NTM	MTB	37	0	0	37
	NTM	0	14	8	22
	Negative	2	12	648	662
	Total	39	26	656	721

## Data Availability

The original contributions presented in the study are included in the article; further inquiries can be directed to the corresponding author.

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
