# Peer review of "Evaluation of the Mycobacterium RealTime Kit Vircell (MRTVircell) Assay for Detecting Mycobacterium Species in Clinical Specimens"

_pathogens, 2025, doi:10.3390/pathogens14050429_

Round 1

Reviewer 1 Report

Comments and Suggestions for Authors

This descriptive but informative study by Aguilera Franco et al. describes the analysis of clinical samples obtained from patients referred for mycobacterial studies to the Microbiology Laboratory of Hospital Universitario Virgen de las Nieves in Andalusia, Spain.  The clinical samples were analyzed using a newly available real time PCR kit (MRTVircell) that can detect both MTBC and NTMs.  The results from this analysis were compared to the results using standard culture detection methods and another real-time PCR assay (Anyplex MTB/NTM).  The study was well designed and the data generated informative.  The limitations of the study were also adequately described.  The tables were easy to read and gave the relevant information.  I do have one suggestion related to their data analysis.   Their analysis separated the results into respiratory vs non-respiratory samples but did not separate based on sub-category (i.e sputum, lavage or aspirate).  It would be interesting to know if there were any differences in sensitivity and specificity when separated into sub-categories.    

Author Response

Comment: This descriptive but informative study by Aguilera Franco et al. describes the analysis of clinical samples obtained from patients referred for mycobacterial studies to the Microbiology Laboratory of Hospital Universitario Virgen de las Nieves in Andalusia, Spain.  The clinical samples were analyzed using a newly available real time PCR kit (MRTVircell) that can detect both MTBC and NTMs.  The results from this analysis were compared to the results using standard culture detection methods and another real-time PCR assay (Anyplex MTB/NTM).  The study was well designed and the data generated informative.  The limitations of the study were also adequately described. The tables were easy to read and gave the relevant information.  I do have one suggestion related to their data analysis. Their analysis separated the results into respiratory vs non-respiratory samples but did not separate based on sub-category (i.e sputum, lavage or aspirate).  It would be interesting to know if there were any differences in sensitivity and specificity when separated into sub-categories.    

Response: First of all, thank you for your comments on our article. We included all samples received during the study period that, based on clinical suspicion, were categorized as respiratory specimens for pulmonary mycobacteriosis or as non-respiratory specimens for suspected extrapulmonary mycobacterial infections. The clinical characteristics of each category, as well as the types of samples used for diagnosis, differ significantly. Within each category, there is also substantial variation in the distribution of specific specimen types, with sputum being by far the most frequent. As a result, sensitivity and specificity calculations for the subcategories would not be directly comparable due to the imbalance in sample sizes and the varying positivity rates observed. A larger sample size would be required to ensure meaningful comparisons.

Reviewer 2 Report

Comments and Suggestions for Authors

The Present manuscript "Evaluation of the Mycobacterium Real Time Kit Vircell (MRTVircell) Assay for detecting Mycobacterium species in clinical specimens" is a modest effort towards a thesis.

However, there are several serious concerns that should be addressed.

  1. line 30, 88, 165 and 240......."NMT" should be "NTM".
  2. The method lacks essential detailing. Sample handling (extraction protocol, operational details etc) should be explicit as a new assay is being presented here which is expected to be parallel to Anyplex.
  3. There is no mention of a clinically healthy control cohort.
  4. Since this is a "Real-time..." assay, readers must know if the assay was done essentially blind-folded...i.e, culture as gold standard is not real-time assay and takes time. 
  5. Line 115: "......stored at 4C till testing......" what is the time laps here?
  6. In results, the reporting must be presented in tabular form for all types of samples....how many of each of sputum/BAL/aspirates (respiratory) and (pleural fluid/ biopsies/  etc were positive or negative...the clinical study is not only about disease/infection positive/negative it is also about right choice of sample.
  7. Authors did not talk about "False Negative" results. 

Author Response

The Present manuscript "Evaluation of the Mycobacterium Real Time Kit Vircell (MRTVircell) Assay for detecting Mycobacterium species in clinical specimens" is a modest effort towards a thesis.

However, there are several serious concerns that should be addressed.

Thank you for your feedback. However, we would like to clarify that this manuscript is only one part of a PhD thesis project designing and evaluating different diagnostic methods for mycobacterial infections. E.g. RT-PCR, sequencing in direct samples etc.

Comments 1: [line 30, 88, 165 and 240......."NMT" should be "NTM"].

Response 1: We have corrected your indication in the text.

Comments 2: The method lacks essential detailing. Sample handling (extraction protocol, operational details etc) should be explicit as a new assay is being presented here which is expected to be parallel to Anyplex.

Response 2: In response to your suggestion, we have expanded the explanation of the new PCR assay (MRTVircell) and provided additional details on the Anyplex method to enhance clarity and improve reader understanding. See in the Material and Methods section from line 152-167 and from line 169-179.

Comments 3: There is no mention of a clinically healthy control cohort.

Response 3: We did not include a cohort of clinically healthy controls because our objective was to compare the performance of two PCR kits against culture under routine diagnostic laboratory conditions. To achieve this, we analyzed only samples routinely received at our center during the study period. We believe this approach more accurately reflects the practical applicability of these diagnostic kits in routine clinical practice.

Comments 4: Since this is a "Real-time..." assay, readers must know if the assay was done essentially blind-folded...i.e, culture as gold standard is not real-time assay and takes time.

Response 4: Mycobacterial culture, although it can take up to six weeks, remains the gold standard for diagnosing mycobacterial infections and serves as the primary reference against which all new diagnostic tools are evaluated. To ensure a comprehensive assessment and accurate calculation of the sensitivity and specificity of the PCR kits, it was essential to await the final culture results, as they provide the most reliable benchmark for diagnosis.

Comments 5: Line 115: "......stored at 4C till testing......" what is the time laps here?

Response 5: Line 131-132: " The aliquots were stored at 4°C until testing, which was conducted within 48 hours of sample processing."

The time lapse for storage at 4°C was up to 48 hours, depending on the workload at the time. We ensured that all samples were tested within this timeframe to maintain sample integrity and minimize potential degradation.

Comments 6: In results, the reporting must be presented in tabular form for all types of samples....how many of each of sputum/BAL/aspirates (respiratory) and (pleural fluid/ biopsies/  etc were positive or negative...the clinical study is not only about disease/infection positive/negative it is also about right choice of sample.

Response 6: We have modified Table 2 by adding sample types.

Specimen

Nº MTB positive (%)

Nº NTM positive (%)

Culture

Anyplex MTB/NTM

MRTVircell

Culture

Anyplex MTB/NTM

MRTVircell

Respiratory

43 (7.68)

36 (6.43)

38 (6.79)

64 (11.43)

20 (3.57)

18 (3.21)

Sputum

39

34

35

54

19

16

Bronchoalveolar lavage

2

0

1

10

1

2

Bronchial aspirates

2

2

2

0

0

0

Non-respiratory

3 (1.86)

1 (0.62)

1 (0.62)

7 (4.35)

2 (1.24)

8 (4.97)

Biopsies

2

1

1

1

0

1

Sterile body fluids

1

0

0

0

0

1

Pleural fluid

-

-

-

2

2

4

Gastric lavages

-

-

-

2

0

0

Urine

-

-

-

1

0

0

Abscesses

-

-

-

1

0

2

Total

46 (6.38)

37 (5.13)

39 (5.41)

71 (9.85)

22 (3.05)

26 (3.61)

Comments 7: Authors did not talk about "False Negative" results.

Response 7: In response to your suggestion, we have incorporated the following text into the discussion, which we believe will significantly enhance this section. See in line 90-99.

Reviewer 3 Report

Comments and Suggestions for Authors

This manuscript provides an insightful and comprehensive evaluation of the MRTVircell PCR assay for detecting Mycobacterium tuberculosis complex (MTBC) and nontuberculous mycobacteria (NTM) in clinical specimens. The study presents strong data on the sensitivity, specificity, and overall performance of the assay compared to the gold standard of culture and the Anyplex MTB/NTM assay. The results are generally well-presented, and the discussion appropriately places the findings within the broader context of molecular diagnostics for mycobacterial infections.

However, the paper could benefit from more detailed statistical explanations, clearer presentation of data, and a stronger emphasis on the clinical relevance of the findings. Additionally, expanding on the limitations and implications of these findings would strengthen the overall impact of the paper.

Introduction Review:

The introduction does a great job of outlining the background on mycobacterial infections, particularly tuberculosis (TB) and nontuberculous mycobacteria (NTM), as well as the importance of accurate diagnostics. However, there are areas where the introduction could be tightened.

While the introduction discusses the global burden of tuberculosis and NTM infections, a clearer connection to the current study’s rationale would improve coherence. For instance, after discussing the challenges of diagnosing these infections, the paper should immediately lead into the need for better diagnostic tools.

The introduction does a great job explaining the various diagnostic methods available, but more detail on the specific limitations of PCR-based methods (compared to culture, AFB stain, etc.) would help clarify why evaluating the MRTVircell is important.

Methods Review:

The methodology is well-structured and provides a comprehensive description of clinical sample collection, laboratory processing, and molecular assays. However, some areas could benefit from additional clarification or minor refinements to improve rigor, reproducibility, and clarity.

The description of the study setting is clear, but it would be useful to specify the inclusion/exclusion criteria for patient samples. Were samples collected from all suspected mycobacterial infections, or were there specific clinical criteria?

Since non-tuberculous mycobacteria (NTM) are not reportable diseases in Spain, it would be helpful to mention how suspected NTM cases were identified or classified.

The description of culture conditions is appropriate, though specifying how contamination was managed (e.g., NaOH-NALC decontamination) would be valuable.

Were MGIT and VersaTREK used simultaneously, or were samples divided between the two systems? If both were used on the same samples, how were discrepancies resolved?

The explanation of the Mycobacterium RealTime PCR Kit Vircell (MRTVircell) is clear. However, it would be useful to include cycle threshold (Ct) cutoffs for positive results.

For both MRTVircell and Anyplex MTB/NTM assays:

Were there any internal controls for DNA extraction efficiency beyond RNAse P?

Were duplicate PCR runs performed to assess reproducibility?

Was there a limit of detection (LOD) study for these assays in the context of this study?

The sample volume used for DNA extraction (600 μL) is noted, but was this volume the same for all sample types, or were adjustments made for certain sample matrices (e.g., pleural fluid vs. sputum)?

The statistical analysis is appropriate, but it would be beneficial to specify confidence intervals for sensitivity, specificity, PPV, and NPV.

The Cohen’s kappa coefficient is an excellent measure of agreement, but mentioning the interpretation criteria (e.g., "substantial agreement" for kappa >0.6) would add clarity.

McNemar’s test is appropriate for comparing sensitivity, but was any adjustment made for multiple comparisons (if applicable)?

Results Review:

The results are presented in a clear and systematic manner. However, there are a few areas where the presentation could be improved:

The table references and figures seem to play an important role in understanding the results, but they are not detailed in the text itself. The results section could benefit from clearer reference to these tables and figures. For example, when mentioning percentages for sensitivity or specificity, it would be more helpful to directly reference the specific table (e.g., "As seen in Table 1A").

The results provide comprehensive statistics for the performance of the MRTVircell and Anyplex assays. However, it would be helpful to explain the implications of these values in the context of clinical diagnostics. For instance, what does the reported sensitivity for MTB detection mean for patient outcomes? What clinical decision-making benefits would be provided by using MRTVircell over Anyplex?

The exclusion of 40 samples due to contamination is important but could be expanded on. Were these excluded samples evenly distributed across MTBC and NTM cases? How might these exclusions affect the generalizability of the results?

While the paper compares both PCR assays to culture results, it would be helpful to more explicitly state the limitations of culture in this context. For example, the time-consuming nature of culture methods may skew clinical relevance, which is why rapid PCR tests are becoming more popular. It would also be helpful to discuss how much of a clinical advantage MRTVircell has over the Anyplex assay in real-world settings.

Discussion Review:

The discussion provides a thorough analysis of the results in comparison with previous studies. However, a stronger conclusion on the clinical implications of using MRTVircell could help solidify the importance of the study. For example, how might the MRTVircell impact TB/NTM treatment regimens or patient outcomes in high-incidence areas? How would it fit into a diagnostic algorithm in such settings?

Author Response

This manuscript provides an insightful and comprehensive evaluation of the MRTVircell PCR assay for detecting Mycobacterium tuberculosis complex (MTBC) and nontuberculous mycobacteria (NTM) in clinical specimens. The study presents strong data on the sensitivity, specificity, and overall performance of the assay compared to the gold standard of culture and the Anyplex MTB/NTM assay. The results are generally well-presented, and the discussion appropriately places the findings within the broader context of molecular diagnostics for mycobacterial infections.

 However, the paper could benefit from more detailed statistical explanations, clearer presentation of data, and a stronger emphasis on the clinical relevance of the findings. Additionally, expanding on the limitations and implications of these findings would strengthen the overall impact of the paper.

 We have carefully read your comments on each section of the article and have taken all your suggestions into account as far as possible. We believe they improve the quality of our work. Thank you very much for your review.

Introduction Review:

The introduction does a great job of outlining the background on mycobacterial infections, particularly tuberculosis (TB) and nontuberculous mycobacteria (NTM), as well as the importance of accurate diagnostics. However, there are areas where the introduction could be tightened.

 While the introduction discusses the global burden of tuberculosis and NTM infections, a clearer connection to the current study’s rationale would improve coherence. For instance, after discussing the challenges of diagnosing these infections, the paper should immediately lead into the need for better diagnostic tools.

 We have introduced the following paragraph in the introduction of the article:

 Line 72-77. Therefore, in recent years, various strategies have been proposed to achieve rapid diagnosis not only of active TB but also of other non-tuberculous mycobacteria involved in disease. These strategies are diverse, focusing on improving conventional techniques and incorporating genotypic, proteomic, and even bacteriophage-based methods. Among these, nucleic acid amplification techniques currently have the most practical utility, although they still present certain limitations when compared to culture [10].

The introduction does a great job explaining the various diagnostic methods available, but more detail on the specific limitations of PCR-based methods (compared to culture, AFB stain, etc.) would help clarify why evaluating the MRTVircell is important.

 Following your recommendations, we have added the following paragraph:

Line 90-99. Despite the advances in molecular diagnostic tools for mycobacteria, these methods do not replace traditional diagnostic tests but rather complement the established diagnostic approach. This is a topic of ongoing debate due to the high cost and the need for specialized laboratories and qualified personnel. Additionally, PCR-based techniques do not differentiate between viable and non-viable microorganisms, which makes them less useful for patients undergoing treatment, as they may detect residual DNA from non-viable organisms. Furthermore, in areas with low prevalence of disease, the positive predictive value of these molecular methods may be limited, potentially leading to false positives. In these contexts, traditional methods such as culture and AFB stain remain essential for confirming the presence of viable organisms and providing accurate diagnostic results.

Methods Review:

The methodology is well-structured and provides a comprehensive description of clinical sample collection, laboratory processing, and molecular assays. However, some areas could benefit from additional clarification or minor refinements to improve rigor, reproducibility, and clarity.

- The description of the study setting is clear, but it would be useful to specify the inclusion/exclusion criteria for patient samples. Were samples collected from all suspected mycobacterial infections, or were there specific clinical criteria?

Following your suggestion we have specified the workflow with the samples included in our article. We hope that with this modification we have responded to your suggestion.

Linea 132-136. Molecular assays were performed either upon request by the clinician responsible for the patient or when our laboratory identified clinical, radiological, or epidemiological factors suggesting that an early molecular diagnosis could be beneficial. This approach ensured that the study population encompassed cases where PCR-based diagnosis was deemed clinically relevant.

- Since non-tuberculous mycobacteria (NTM) are not reportable diseases in Spain, it would be helpful to mention how suspected NTM cases were identified or classified.

In the vast majority of cases, the symptoms are almost indistinguishable from those caused by MTB, except in specific cases such as immunocompromised patients (e.g., HIV and M. avium infections) or certain extrapulmonary sites where particular species of non-tuberculous mycobacteria may be more prevalent. As a result, there is often no clear initial suspicion of NTM infection. In light of this, we believe the previously added paragraph outlining the sample selection criteria adequately addresses this point.

- The description of culture conditions is appropriate, though specifying how contamination was managed (e.g., NaOH-NALC decontamination) would be valuable.

The Materials and Methods section references the decontamination methods used for non-sterile samples, along with the corresponding citations that describe these methodologies in detail.

Line 128-129: "Cultures were maintained for six weeks following the decontamination of non-sterile samples, in accordance with standard protocols [6, 16]."

  1. Forbes BA, Hall GS, Miller MB, Novak SM, Rowlinson MC, Salfinger M, et al. Practice guidelines for clinical microbiology laboratories: mycobacteria. Clin Microbiol Rev. 2018;31(2):e00038-17. doi:10.1128/CMR.00038-17.
  2. Kubica GP, Dye WE, Cohn ML, Middlebrook G. Sputum digestion and decontamination with N-acetyl-L-cysteine-sodium hydroxide for culture of mycobacteria. Am Rev Respir Dis. 1963 May;87:775-9. doi:10.1164/arrd.1963.87.5.775.

Given that these are routine procedures widely used in all mycobacteriology laboratories, we do not consider a detailed description necessary.

- Were MGIT and VersaTREK used simultaneously, or were samples divided between the two systems? If both were used on the same samples, how were discrepancies resolved?

In our laboratory, we have two liquid culture systems (MGIT and VersaTREK), both widely used for mycobacterial culture in routine clinical laboratories. Samples were plated in only one of the two systems based on availability; therefore, they were not plated in parallel. As a result, there was no need to resolve discrepancies, as we only obtained a single culture result per sample.

- The explanation of the Mycobacterium RealTime PCR Kit Vircell (MRTVircell) is clear. However, it would be useful to include cycle threshold (Ct) cutoffs for positive results.

Thank you very much for the suggestion, we have added this paragraph to the Material and Methods section:

Line 165-167. The threshold cycle (Ct) values used to determine positive results in the MRTVircell assay were <37 for NTM and <40 for MTBC, MAC, and MABS, according to the manufacturer's recommendations.

For both MRTVircell and Anyplex MTB/NTM assays:

- Were there any internal controls for DNA extraction efficiency beyond RNAse P?

The MRTVircell kit includes only RNAse P as an internal control, which is sufficient to verify the proper extraction of the sample, confirm the absence of amplification inhibitors, and ensure correct amplification setup. In contrast, the Anyplex MTB/NTM kit contains plasmid DNA as an internal control, designed to guarantee equivalent amplification of the internal control, MTB, and mycobacterial target DNA. However, it does not include a specific control to confirm the correct extraction process of the sample.

We had not previously explained the presence of an internal control in the Anyplex MTB/NTM kit because it has been commercially available for years. Instead, we focused on describing the MRTVircell kit as it is a new diagnostic tool. However, in response to your suggestion, we have now included this information.

We have added the following in the text:

Line 149-151: As a control procedure, the kit amplifies the human RNAse P gene as an internal control to ensure proper sample extraction, the absence of amplification inhibitors, and the correct assay setup.

Line 175-177: Each run included positive and negative controls to ensure the accuracy of the amplification process, along with plasmid DNA as an internal control to verify consistent amplification of the internal control, MTBC, and mycobacterial target DNA.

- Were duplicate PCR runs performed to assess reproducibility?

No, duplicate PCR runs were not performed to assess reproducibility. Each sample was tested once with each assay, following the manufacturer's instructions.

- Was there a limit of detection (LOD) study for these assays in the context of this study?

We did not perform a limit of detection (LOD) study as part of this research, as this evaluation was conducted by the manufacturer during the development of the assay. The LOD data are provided in the kit insert, which contains detailed information on the assay's analytical performance.

- The sample volume used for DNA extraction (600 μL) is noted, but was this volume the same for all sample types, or were adjustments made for certain sample matrices (e.g., pleural fluid vs. sputum)?

Yes, that was the volume used for all samples.

- The statistical analysis is appropriate, but it would be beneficial to specify confidence intervals for sensitivity, specificity, PPV, and NPV.

In response to your suggestion, we have added confidence intervals to the tables 1A and 1B.

Target

AFB smear

Assays

Culture + (n=117)

Culture - (564)

Sensitivity / Specificity

PPV / NPV

PCR +

PCR -

PCR +

PCR -

MTB

Positive

Anyplex MTB/NTM

25

0

1

8

100/88.89 [68.36 - 109.42]

96.15 [89.09 - 100]/100

MRTVircell

25

0

1

8

100/88.89 [68.36 - 109.42]

96.15 [89.09 - 100]/100

Negative

Anyplex MTB/NTM

10

11

1

549

47.62 [26.83 - 69.41]/99.82 [99.16 - 99.99]

90.91 [62.26 - 98.38]/98.04 [96.63 - 98.92]

MRTVircell

12

9

1

550

57.14 [34.85 - 76.81]/99.82 [99.16 - 99.99]

92.31 [66.74 - 98.63]/98.39 [97.06 - 99.16]

All

Anyplex MTB/NTM

35

11

2

557

76.09 [61.23 - 86.75]/99.64 [98.8 - 99.91]

94.59 [81.81 - 98.6]/98.06 [96.63 - 98.92]

MRTVircell

37

9

2

558

80.43 [65.7 - 89.88]/99.64 [98.81 - 99.91]

94.87 [82.7 - 98.64]/98.41 [97.06 - 99.16]

Table 1A. Performance of real-time PCR systems analyzed based on AFB smear and culture results for MTB

Target

AFB smear

Assays

Culture + (n=117)

Culture - (564)

Sensitivity / Specificity

PPV / NPV

PCR +

PCR -

PCR +

PCR -

NTM

Positive

Anyplex MTB/NTM

5

1

0

8

83.33 [62.23 - 100]/100

100/88.89 [70.84 - 100]

MRTVircell

5

1

0

8

83.33 [62.23 - 100]/100

100/88.89 [70.84 - 100]

Negative

Anyplex MTB/NTM

10

55

5

549

15.38 [8.21 - 26.33]/99.1 [97.94 - 99.65]

66.67 [35.42% - 88.72]/90.89 [88.34 - 92.93]

MRTVircell

15

50

4

550

23.08 [14.07 - 35.02]/99.28 [98.19 - 99.73]

78.95 [54.43 - 92.86]/91.67 [89.26 - 93.59]

All

Anyplex MTB/NTM

15

56

5

557

21.13 [12.93 - 32.22]/99.11 [97.95 - 99.65]

75 [50.9% - 89.87]/90.86 [88.31 - 92.91]

MRTVircell

20

51

4

558

28.17 [18.63 - 40.13]/99.29 [98.2 - 99.73]

83.33 [62.62 - 93.98]/91.63 [89.22 - 93.56]

Table 1B. Performance of real-time PCR systems analyzed based on AFB smear and culture results for NTM

- The Cohen’s kappa coefficient is an excellent measure of agreement, but mentioning the interpretation criteria (e.g., "substantial agreement" for kappa >0.6) would add clarity.

Modifications in the text:

Line 233-25. According to culture findings, the overall concordance rate between the two PCR methods was 88.11% (600/681), with a Cohen’s kappa (κ) of 0.61 for MRTVircell, indicating substantial agreement, and 0.55 for Anyplex MTB/NTM, indicating moderate agreement.

Line 245-247. The overall concordance rate between the two PCR methods, regardless of culture results, was 96.95% (699/721), with a Cohen’s kappa (κ) of 0.806, indicating almost perfect agreement (see Table 4).

-  McNemar’s test is appropriate for comparing sensitivity, but was any adjustment made for multiple comparisons (if applicable)?

McNemar’s test was used to compare both the sensitivity and specificity of the two PCR kits. Since only two pre-specified comparisons were performed, a correction for multiple comparisons was not applied. The sensitivity, specificity, PPV, and NPV values were reported as descriptive statistics.

Results Review:

The results are presented in a clear and systematic manner. However, there are a few areas where the presentation could be improved:

- The table references and figures seem to play an important role in understanding the results, but they are not detailed in the text itself. The results section could benefit from clearer reference to these tables and figures. For example, when mentioning percentages for sensitivity or specificity, it would be more helpful to directly reference the specific table (e.g., "As seen in Table 1A").

We have revised the Results section as directed to include explicit references to the corresponding tables where the data are presented. Line 210, 214, 216, 218, 222, 226, 229, 232, 239, 244.

- The results provide comprehensive statistics for the performance of the MRTVircell and Anyplex assays. However, it would be helpful to explain the implications of these values in the context of clinical diagnostics. For instance, what does the reported sensitivity for MTB detection mean for patient outcomes? What clinical decision-making benefits would be provided by using MRTVircell over Anyplex?

We have added the following paragraph in line 301-308:

Early diagnosis of TB is essential for timely treatment and infection control. Rapid detection of MTBC and its differentiation from NTM help prevent disease transmission and ensure appropriate treatment. In this study, the MRTVircell and Anyplex assays demonstrated high sensitivity for MTBC detection (80.43% and 76.09%, respectively), with MRTVircell showing slightly superior performance. This increased sensitivity may offer clinical advantages, particularly in settings where rapid and accurate diagnosis is critical for patient management and public health interventions.

The ability to differentiate between NTM species and their benefits when making clinical decisions we think is explained in the text: line 336-345. Included in the discussion of the first version of the article.

- The exclusion of 40 samples due to contamination is important but could be expanded on. Were these excluded samples evenly distributed across MTBC and NTM cases? How might these exclusions affect the generalizability of the results?

Thank you for your insightful comment. We appreciate your suggestion to further elaborate on the exclusion of the 40 samples due to contamination. Among these, 38 samples returned negative results for both PCR assays, while 2 tested positive for NTM with both MRTVircell and Anyplex MTB/NTM. Since no MTBC cases were detected among the excluded samples, we do not expect these exclusions to have significantly influenced the reported sensitivity and specificity for MTBC detection. Additionally, given that both assays yielded concordant results for the two NTM-positive samples, their exclusion is unlikely to have introduced a bias favoring one method over the other. However, we recognize that sample contamination is an inherent limitation in mycobacterial culture, and we have added a brief discussion on this point in the manuscript.

Following your advice we have added the following paragraph in Results.

Line 200-206. A total of 40 samples (5.5%) were excluded from the sensitivity and specificity evaluation due to culture contaminated by bacteria or fungi. Among these, both PCR tests yielded negative results in 38 out of 40 cases (95%), while 2 samples (5%) tested positive for NTM with both MRVircell and Anyplex MTB/NTM. Since no MTBC cases were affected, and the two NTM-positive cases yielded concordant results, these exclusions are unlikely to have significantly impacted the reported sensitivity and specificity values. Nonetheless, culture contamination remains an inherent limitation in mycobacterial diagnostics.

- While the paper compares both PCR assays to culture results, it would be helpful to more explicitly state the limitations of culture in this context. For example, the time-consuming nature of culture methods may skew clinical relevance, which is why rapid PCR tests are becoming more popular. It would also be helpful to discuss how much of a clinical advantage MRTVircell has over the Anyplex assay in real-world settings.

We have modified the following paragraph in the article.

Line 380-387: The MRTVircell is a CE-IVDR-marked assay for the direct detection and differentiation of Mycobacterium spp., MTBC, MAC, and MABC. It is provided as a lyophilized, ready-to-use master mix, offering a rapid diagnostic alternative to culture, which can take up to six weeks. The assay provides reliable results in less than 3.5 hours, enabling timely clinical decision-making and potentially reducing transmission risks. In comparison to the Anyplex assay, MRTVircell differentiates MAC and MABC from other NTM species, which is crucial for guiding appropriate treatment strategies, as MAC and MABC infections require targeted therapies.

Discussion Review:

- The discussion provides a thorough analysis of the results in comparison with previous studies. However, a stronger conclusion on the clinical implications of using MRTVircell could help solidify the importance of the study. For example, how might the MRTVircell impact TB/NTM treatment regimens or patient outcomes in high-incidence areas? How would it fit into a diagnostic algorithm in such settings?

We have expanded the conclusions following their recommendations.

Line 388-393. In conclusion, this study demonstrates that in a setting of low TB incidence and with the increasing prevalence of NTM infections, especially respiratory, the MRTVircell assay is a reliable and effective tool for the detection and differentiation of mycobacterial infections. By performing molecular testing with multiple mycobacterial targets, it enables the establishment of effective treatment from the outset, improving clinical management and patient outcomes.

Round 2

Reviewer 2 Report

Comments and Suggestions for Authors

The resubmitted manuscript still lacks clear statement if the samples were tested blind-folded. We know that Culture is gold standard but not absolutely accurate either, particularly in paucibacillary situations. Therefore, stating the mismatched diagnosis should give an honest assessment for the proposed kit. Also, lack of healthy control really a limiting situation for this study.